# Chemical Profile, Antioxidant and Anti-Inflammatory Potency of Extracts of *Vitex madiensis* Oliv. and *Crossopteryx febrifuga* (Afzel ex G. Don)

**DOI:** 10.3390/plants12020386

**Published:** 2023-01-13

**Authors:** Ghislaine Boungou-Tsona, Maël Gainche, Caroline Decombat, Isabelle Ripoche, Kevin Bikindou, Laetitia Delort, Florence Caldefie-Chézet, Aubin Loumouamou, Pierre Chalard

**Affiliations:** 1Equipe Pluridisciplinaire de Recherche en Alimentation et Nutrition (EPRAN), Faculté des Sciences et Techniques, Université Marien Ngouabi, Brazzaville BP 389, Congo; 2Département des Sciences Chimiques, Institut National de Recherche en Sciences Exactes et Naturelles (IRSEN), UR Chimie des Substances Naturelles, Cité Scientifique de Brazzaville, Brazzaville BP 2400, Congo; 3Institut de Chimie de Clermont-Ferrand, Université Clermont Auvergne, Clermont Auvergne INP, Centre National de la Recherche Scientifique, F-63000 Clermont-Ferrand, France; 4Unité de Nutrition Humaine, l’Alimentation et l’Environnement, Institut National de Recherche pour l’Agriculture, Université Clermont-Auvergne, CRNH Auvergne, F-63000 Clermont-Ferrand, France

**Keywords:** chemical profiles, African plants, antixodant and anti-inflammatory activities

## Abstract

*Vitex madiensis* Oliv. (Lamiaceae) and *Crossopteryx febrifuga* (Rubiaceae), two plants commonly used in traditional African medicines to treat malaria and pain, were studied either to determine their chemical profiles or to evaluate their antioxidant and anti-inflammatory activities. In this study, we investigated leaves, trunk bark, root bark and fruits methanolic extracts of both plants in order to find out which part of the plant is responsible for the activity. The analyses of the chemical profiles allowed us to confirm the presence of several ecdysteroids, especially 20-hydroxyecdysone in some parts of *V. madiensis* and to highlight the presence of organic acids and phenol derivatives in *C. febrifuga*. Among the four parts of the plants studied, only the fruits extract of *C. febrifuga* could present anti-inflammatory activity by decreasing ROS production. The leaves and trunk bark extracts of *V. madiensis* showed significant free radical scavenging activity compared to ascorbic acid, and the same extracts decrease ROS production significantly. The activity of these two extracts could be explained by the presence of ecdysteroids and flavonoids. The ROS production inhibition of *V. madiensis* is particularly interesting to investigate with further analyses.

## 1. Introduction

*Crossopteryx febrifuga* (Rubiaceae) and *Vitex madiensis* Oliv. (Lamiaceae) are two species growing in tropical and subtropical regions throughout the world. These plants are widely present in the savannah areas of West, East and Central Africa [1,2,3]. In Congo-Brazzaville, these two species grow particularly in the area of the “Plateau des cataractes” (Pool department) in the southern part of the country. These two plants are commonly advised for the treatment of pain by traditional healers.

*C. febrifuga* is a shrub whose various organs have multiple uses in traditional African medicine. Indeed, this plant has been used for many years for the treatment of pain and malaria, and its effectiveness is widely acclaimed, particularly by the Hausa communities of Nigeria [4]. The fruits are used in Mali for treatment of respiratory tract infections, as well as an antitussive and as a febrifuge [5]. The bark is used to treat fever and diarrhea in Guinea [6]. The roots, leaves and trunk bark are used to relieve stomach aches, chest pain and as an analgesic [7]. In Congo-Brazzaville, the fruit is used to treat diarrhea and fever. In the same country, this plant is used to treat headaches, migraine, bacterial infections and epilepsy [8]. In addition, several studies have demonstrated that methanolic extracts of different parts of the plant have antimalarial, antipyretic, analgesic, antioxidant and anti-inflammatory properties [1,4,9]. Despite these uses in traditional medicine and the evaluation of biological properties of crude extracts, the phytochemical profile of this plant and its relationship with therapeutic uses have not been extensively investigated. Two bisdesmoside saponins were isolated [10] from *C. febrifuga* roots and two alkaloids, crossopterin and crossoptin, have been isolated from bark extracts [11].

*V. madiensis* is a shrub that is about 5 m tall with a massive underground woody rootstock. The leaves, fruits, stem bark and roots have traditional medicinal uses in several African countries [12]. The plant is used for the treatment of headaches, toothaches, aches and pains and more generally as an analgesic. In Gabon and Angola, it is used to treat the symptoms of malaria [13,14,15]. Unlike other species of the genus *V. madiensis*, there are few phytochemical studies on *V. madiensis*. These studies mentioned the identification of phytoecdysteroids from the root bark [16], particularly 20-hydroxyecdysone and ajugasterone C. Phytoecdysteroids have important physiological effects on insects, particularly in the defense of plants against them [17]. In general, species of the *Vitex* genus contain a variety of potentially bioactive molecules, such as iridoids, flavonoids, diterpenoids and phytosteroids [18,19], thus explaining the origin of the observed biological properties.

The objective of this work was to carry out a comparative study of the chemical profiles of methanolic extracts of the different parts (leaves, fruits, trunk bark and root bark) of *C. febrifuga* and *V. madiensis* and to determine their antioxidant properties by evaluating their free radical scavenging activity and their anti-inflammatory activity with tests on reactive oxygen species (ROS) production by leukocytes.

## 2. Results

### 2.1. Chemical Profiles

A methanolic maceration of each part of plant was carried out to extract most of the secondary mebabolites. By comparing the yields obtained (Table 1), we could conclude that the yields were quite similar for all parts of *V. madiensis* (between 3 and 6%). For *C. febrifuga*, the yields were quite high for the leaves, trunk bark and roots (between 10 and 19%), but the yield is lower for fruits (3%).

The extract of each part of plants was analyzed by HPLC in order to determine the best conditions to separate most of the compounds present in the extracts. For both plants, the analyses were performed using a method developed in our group to study complex matrices and for which a database of around one hundred compounds was established using commercially available standards.

#### 2.1.1. Extracts of *V. madiensis*

To determine the chemical composition of each extract, we carried out LC/MS analyses either in positive and negative mode to identify the compounds present in the extract using the method developed by HPLC. As the positive mode did not give more information about the chemical profile, we only investigated the negative mode. The chromatographic profiles of the extracts of leaves, trunk bark, root bark and fruits of *V. madiensis* are presented in Figure 1.

These results showed that the main compounds present in both extracts were ecdysteroids, considered as chemotaxonomic markers of the genus *Vitex* [32]. Among the family of ecdysteroids, we identified four major compounds: 20-hydroxyecdysone 10, isovitexirone 14, vitexirone 15 and ajugasterone C 16 (Figure 2).

Other families of compounds have also been identified such as organic acid, flavonoids and fatty acids. Indeed, leaves and trunk bark extracts contain flavone glycosides such as homoorientin, orientin and vitexin; and flavonoids such as luteolin, 3,7-dimethylquercetin. The presence of caffeic acid derivatives such as dicaffeoylquinic acid were detected in both extracts (Table 1).

Comparison of the four chromatograms allowed us to conclude that the leaves extract contain more families of compounds (ecysteroids, flavonoids, fatty acids) than the other parts. Root bark and fruits contain mainly ecdysteroids and the major compound of both extracts seemed to be 20-hydroxyecdysone. As the four LC/MS analyses were performed with the same concentration of extract, we could also compare the relative quantity of ecdysteroids, potentially responsible for the biological activities of each extract. We could estimate that the root bark extract is more concentrated in ecdysteroids than the other parts of the plant, and the leaves and fruits extracts seem to be rich in 20-hydroxyecdysone.

#### 2.1.2. Extracts of *C. febrifuga*

As previously described, chemical profiles of extracts from leaves, trunk bark, root and fruit of *C. febrifuga* were performed by LC/MS analysis (Figure 3).

The chromatograms showed that the extracts of the different parts of *C. febrifuga* were quite similar events if some compounds were not present in the same concentration. The extracts contained especially organic acids, phenols, particularly chlorogenic acid derivatives, and flavonoids (quercetin derivatives). Organic acids such as citric acid, protocatechic acid and vanillic acid are present in the fruits extract. The other extracts contain mainly iridoid glycosides such as geniposidic acid, loganin, 11-methylixoside and glycosylated flavonoides such as hyperoside (Figure 4). The trunk barks contain geniposidic acid and quinic acid esters such as chlorogenic acid; the root bark extract contains caffeic acid diester, 3,4-dicaffeoylquinic acid and shanzhiside. The whole extracts contain the same unidentified compound possessing a retention time of 12.2 min. The structure of this compound has not been yet identified as it has not been described in *Crossopteryx* species. The identified compounds and their masses are presented in Table 2.

### 2.2. Antioxidant Activity of Extracts

The antioxidant potency of the methanolic extracts was evaluated by measuring their free radical scavenging activity using a DPPH assay. The IC_50_ (concentration providing 50% inhibition) was calculated graphically using a calibration curve in the linear range by plotting the extract concentration vs. the corresponding scavenging effect. The extracts from the leaves, stem barks and fruits of *V. madiensis* significantly inhibit the DPPH radical. The inhibitory concentrations (IC_50_) are 110, 125 and 210 µg/mL, respectively. The leaves and trunk bark showed promising antioxidant potential with IC_50_ close to that of ascorbic acid, taken as the reference antioxidant (IC_50_ = 100 µg/mL) (Table 3). These results are comparable with those obtained with other *Vitex species*, in particular *Vitex doniana*, *Vitex fischeri* and *Vitex kiniensis* [33,34].

**Table 2 plants-12-00386-t002:** Compounds identified in extracts of leaves, trunk bark, root bark and fruits of *Crossopteryx febrifuga*.

N°	Composés	Tr (min)	Formula	M-H	MS^2^ (m/z)	L	TB	RB	F	Reference
1	Uronic acid	3.34	C_6_H_12_O_7_	195.0499	75/**195**/129/87/99/89/85	+	+	++	+	Standard
2	Quinic acid	3.87	C_7_H_12_O_6_	191.0547	**191**/85/127/93/85	+++	+++	+	+++	Standard
3	Sucrose	4.11	C_12_H_22_O_11_	341.1087		-	+	+	+	Standard
4	Citric acid	6.80	C_6_H_8_O_7_	191.0186	**191**/129/111/87/85	+	-	+	+	Standard
5	Fumaric acid	7.34	C_4_H_4_O_4_	115.0032	71/**115**/72/51	-	-	-	+	[35]
6	Shanzhiside	9.75	C_16_H_24_O_11_	391.1241	183/165/139/99/89/71/101/**391**	-	-	+	-	[36]
7	Geniposidic acid	10.03	C_16_H_22_O_10_	373.1135	123/149/211/167/**373**/193	+	+	-	-	[37]
8	Ixoside	10.40	C_16_H_20_O_11_	387.0927	181/93/343/89/59/137/163/119/71/101/205/**387**	-	+	++	-	[38]
9	Protocatechuic acid	11.43	C_7_H_6_O_4_	153.0177	109/**153**/110	-	-	-	+	[22]
10	Shanzhiside methyl ester	12.23	C_18_H_28_O_13_	451.1454 *	243/101/405/**451**	+	+++	+++	-	[39]
11	Chlorogenic acid	12.98	C_16_H_18_O_9_	353.0878		-	+	-	+	Standard
12	Rehmannioside A	13.47	C_21_H_32_O_15_	523.1664	293/89/233/71/125/477	-	-	+	-	[37]
13	4-Hydroxybenzoic acid	13.80	C_7_H_6_O_3_	137.0238	**137**/138/136/109/108/81/119/93	-	-	-	+	[22]
14	Loganine	14.69	C_18_H_28_O_12_	435.1504 *	227/101/139/**435**	+	-	-	-	[40]
15	11-Methylixoside	15.03	C_17_H_22_O_11_	401.1086	101/137/**401**/195/93/239/221	-	+	++	-	[41]
16 17	Hyperoside Isoquercetin	18.10 18.51	C_21_H_20_O_12_ C_21_H_20_O_12_	463.0881 463.0881	300/**463**/271/255/179 300/**463**/271/255/179/151	+ +	- -	- -	+ -	Standard Standard
18	Vanillic acid	19.78	C_8_H_8_O_4_	167.0338	167/152/111	-	-	-	+	Standard
19	3,4-dicaffeoylquinic acid	24.89	C_25_H_24_O_12_	515.1189		-	-	+	-	Standard
20	Azelaic acid	26.71	C_9_H_16_O_4_	187.0965	125/187/169/126/97/143	-	-	-	+	Standard

+: presence of the compound in the extract; -: absence of the compound in the extract; L: leaves, TB: trunk bark; R: root bark; F: fruit; * (M+HCOO^−^), fragmented ions are highlighted in bold.

Similarly for *C. febrifuga*, extracts from the leaves, trunk bark and root bark significantly inhibited the DPPH radical (Table 3), with IC_50_s of 100, 110 and 200 µg/mL, respectively. Fruit extract did not have a significant effect (IC_50_ = 710 µg/mL).

### 2.3. ROS Production Inhibition

In order to investigate further the anti-inflammatory and antioxidant effects of both plants, we examined the ability of the extracts to decrease cellular ROS production by incubating blood leukocytes from healthy donors stimulated by PMA. Stimulation with PMA, a direct PKC activator, resulted in a significant increase in ROS production after 2 h of incubation. Different concentrations of each extract of *V. madiensis* and *C. febrifuga* were tested (0 (Control), 10, 25, 50, and 100 µg/mL). Concerning the *V. madiensis* extracts, the leaf extract seemed most active reducing ROS production by 27 to 58% with increasing concentration, and this inhibition was statistically significant from a concentration of 25 µg/mL (Figure 5). The trunk bark and fruit extracts also reduced the production of ROS induced by PMA stimulation of leukocytes to become significant at concentrations of 50 and 100 µg/mL, respectively. The effect of the bark extract appeared to be dose-dependent at the lowest concentrations (R^2^ = 0.9979). For root bark extract, there was a large inter-individual variability; nevertheless, ROS production appeared to decrease in a dose-dependent manner (R^2^ = 0.9915).

With regard to *C. febrifuga*, results showed that none of the extracts had a real effect on the production of ROS of leukocytes stimulated by PMA, except for the fruit extract, which significantly inhibited this production compared to the control from 50 µg/mL until an effect of around 42% was reached at 100 µg/mL (Figure 6).

### 2.4. Effect of Extracts on Leukocyte Viability

We then examined the impact of the extracts on leukocyte viability in order to address the issue that decreased ROS production was related to decreased viability. This study was carried out under the same conditions as above, but this time over a period of 24 h using the resazurin test. It was shown that the leaves and trunk bark extracts of *Vitex madiensis* (Figure 7) as well as the fruit and root bark extracts of *Crossopteryx febrifuga* (Figure 8) did not show any effect on cell viability. No significant difference was observed for these extracts (*p* > 0.05) at all the concentrations tested compared to the 100% normalized Control (cells incubated with PMA and without extract). The other extracts showed effects on cell viability at 50 and 100 µg/mL at 24 h (*p* < 0.05).

## 3. Discussion

The chemical profiles of the methanolic extracts of various parts of *V. madiensis* were investigated and allowed us to identify four phytosterols: 20-hydroxyecdysone, ajugasterone C, isovitexirone and vitexirone. These ecdysteroids, found in some species of the *Vitex* genus, were identified in leaves, trunk bark and root bark, and fruits, confirming the hypothesis that ecdysteroids are taxonomic markers of the *Vitex* genus [32]. As previously described, 20-hydroxyecdysone is the most present ecdysteroids in *Vitex* species [32] and has been isolated from *V.* madiensis [16]. Ajugasterone C and vitexirone have been identified in *Vitex fisherii* [28], *Vitex leptobotrys* [42], *Vitex polygama* [43], *Vitex scabra* [44] and *V. madiensis.* Moreover, ecdysteroids, and especially 20-hydroxyecdysone, possess various biological activities such as anti-diabetic, antioxidant, anti-inflammatory, angiogenic, cardioprotective, neuroprotective, lung, kidney and gastric proactive activities. A recent study demonstrated that 20-hydroxyecdysone could prevent the appearance of severe forms of COVID-19 [45].

The extracts of *V. madiensis* showed a significant free radical scavenging activity compared to ascorbic acid. The leaves and trunk bark extracts possess the highest activity, confirming the antioxidant power of *V. madiensis*, as already observed in previous studies 19. It should also be noted that many species of the *Vitex* genus, especially leaf and fruit extracts, show significant scavenging activity [16,19,33,46].

To our knowledge, the activity of *V. madiensis* extracts on the ROS production induced by blood leukocytes has not been described, and we found it relevant to explore this field. First, it was established that leaf extract at 25 µg/mL reduced ROS production significantly after 2 h of incubation with stimulated leukocytes by PMA, without affecting the viability of cells. Our study also showed that the extracts of trunk bark and fruits of *V. madiensis* had an effect on ROS production by leukocytes, and it became significant at concentrations of 50 and 100 µg/mL, respectively. Even then, the viability study of leukocytes confirmed the antioxidant effect of *Vitex madiensis* extracts as it was not the consequence of a decrease in leukocyte viability. These results confer to the extracts both an anti-radical effect and a potential anti-inflammatory activity. Oxidative species are produced not only under pathological situations (inflammatory diseases, cancers, autoimmune diseases, etc…) but also during physiological situations such as cellular metabolism [47,48,49]. Indeed, in the body, ROS are chemical mediators that are involved in cellular communication and are part of the defence mechanisms in an inflammatory context [50,51,52]. On the other hand, if the ROS production exceeds the defence capacity, they are able to induce direct cellular injury via oxidative stress to nucleic acids, cellular proteins and by activating lipid peroxidation, which destroys the cell membrane [31,53,54]. Thus, the neutralization of free radicals or reactive oxygen species by a naturally antioxidant extract is valuable for cell protection from oxidative stress.

The antioxidant activity and the ROS production inhibition of *V. madiensis* extracts could be due to the presence of 20-hydroxyecdysone and probably the ecdysteroids in high concentration in the plant. Indeed, ecdysteroids have already been described for their antioxidant capacity. In particular, ecdysterone exerts protective effects against lipid peroxidation from free radicals, obtaining a status of antioxidant [55,56].

The methanolic extracts of *C. febrifuga* contained mostly iridoids derivatives and flavonoids. The trunk bark and the root bark extracts were particularly rich in glycosilated iridoids such as ixoside, shanziside and methyl ester shanziside, which seem to be the major compounds of these two extracts. The fruits extract contained widely flavonoids and phenolic derivatives such as hyperoside and chlorogenic acid. The extracts of leaves, trunk bark and root bark of *C. febrifuga* showed significant antioxidant effects due to their free radical scavenging activity. The extracts of leaves and trunk bark present a higher antioxidant capacity compared to the root extract with an IC_50_ close to ascorbic acid. In the literature, several authors state that the antioxidant activity of plant extracts depends on its content of phenolic compounds [33,57,58]. In addition, flavonoids are the most important class of polyphenols that have shown greater potential for biological activities such as antioxidant, anti-inflammatory, antibacterial, anticancer and anti-allergic activity [59]. This partly coincides with our study, whose LC-MS profiles of the different extracts of *C. febrifuga*, and particularly the fruit extract, are characterized by a few phenolic and flavonoid compounds, which would be responsible for the antioxidant potential observed with DPPH.

The fruit extract did not show a significant effect on the DPPH radical, unlike previous studies carried out on the fruits of this plant whose results showed significant effects on the DPPH radical [60]. However, this fruit extract inhibited ROS production induced by blood leukocytes stimulated with PMA and this, significantly at concentrations above 50 µg/mL. It would be interesting to investigate which constituent could be involved, at least in part, in the protective effect against ROS toxicity.

## 4. Materials and Methods

### 4.1. Plant Material

The plant material (leaves, fruits, trunk bark and root bark) of *Vitex madiensis* Oliv. and *Crossopteryx febrifuga* (Afzel ex G. Don) BENTH was collected in the savannah zone, 25 km South of Brazzaville. After collection of the samples, one specimen of each plant species was authenticated and preserved in the herbarium of the Institut National de Recherche en Sciences Exactes et Naturelles (IRSEN) of Brazzaville under the number T7041 for *Vitex madiensis* Oliv. and B434 for *Crossopteryx febrifuga* (Afzel ex G. Don) BENTH. The plant material was dried in the shade for 7 days and then ground with a grinder.

### 4.2. Cell Material

Blood (collected from healthy volunteers, *n* = 3) was provided by the Etablissement Français du Sang (EFS) of the city of Clermont-Ferrand according to the contract n°16-21-62 (in accordance with the following articles L1222-1, L1222-8, L1243-4 and R1243-61 of the public health code). Whole blood leukocytes were obtained as previously described by Cholet et al. [61].

### 4.3. Methanolic Extracts

All the parts of the plants were air-dried grinded and then were macerated in methanol for 24 h (plant/solvent ratio: 1/10). In each case, the mixture was then filtered and evaporated under vacuo to dryness. The yields obtained for each part of the plant are summarized in Table 4. The crude extract of each plant part is obtained and then stored at 4 °C before analysis.

### 4.4. Determination of the Chemical Profiles

The chemical profile of each crude extract was determined by chromatographic analyses using Ultra-High Performance Liquid Chromatography (UHPLC). HPLC-MS analyses were performed on an Ultimate 3000 RSLC UHPLC system (Thermo Fisher Scientific Inc., Waltham, MA, USA) coupled to a quaternary rapid separation pump (Ultimate autosampler) and a rapid separation diode array detector. Compounds were separated on an Uptisphere Strategy C18 column (250 × 4.6 mm, 5 µm, Interchim, Montluçon, France), which was controlled at 30 °C. The mobile phase was a mixture of 0.1% (*v*/*v*) formic acid in water (phase A) and 0.1% (*v*/*v*) formic acid in acetonitrile (phase B). The gradient of phase A was 100% (0 min), 80% (10 min), 73% (35 min), 0% (40–50 min) and 100% (51–60 min). The flow rate was 0.8 mL/min, and the injection volume was 5 µL. The UHPLC system was connected to an Orbitrap (Thermo Fisher Scientific Inc., Waltham, MA, USA) mass spectrometer, operated in the positive and negative electrospray ionization mode. Source operating conditions were: 3 kV spray voltage; 320 °C heated capillary temperature; 400 °C auxiliary gas temperature; sheath, sweep and auxiliary gas (nitrogen) flow rate 50, 10 and 2 arbitrary units, respectively; and collision cell voltage between 10 and 50 eV. Full scan data were obtained at a resolution of 70,000, whereas MS^2^ data were obtained at a resolution of 17,500. Data were processed using Xcalibur software (Thermo Fisher Scientific Inc., City, MA, USA).

The sample to be analysed was prepared under the following operating conditions: 5 mg of the methanolic extract was diluted in 5 mL of MeOH of HPLC grade quality. The solution is then filtered with a 0.45 μm PTFE filter. Part of the filtrate is placed in a vial for analysis.

### 4.5. Antioxidant Properties

The in vitro antiradical activity was evaluated by the 2,2 diphenyl-1-picrylhydrazyl radical (DPPH) scavenging method. The protocol, previously described in literature, was used with some modifications [62,63].

Extract and ascorbic acid solutions in concentrations ranging from 200 to 1200 µg/mL, and 0.18 mM DPPH were prepared in methanol. In addition, 300 µL of each concentration of extracts and ascorbic acid were mixed with 3 mL of 0.18 mM DPPH solution. The kinetics were monitored for 45 min at 517 nm using a visible Jasco UV spectrophotometer (a methanolic solution was used as blank). The experiments were repeated three times, and the results were presented as a mean ± SEM. The inhibitory concentration 50 (IC_50_) of each extract was determined. The lower the IC_50_ value, the better the antiradical activity [64].

### 4.6. ROS Production Inhibition

The ROS production inhibition was determined by the method of inhibiting the production of reactive oxygen species (ROS) induced by human blood leukocytes.

The leukocyte preparations (*n* = 3) were obtained as previously described. The cells were placed in a 96 well polystyrene plate (Cell Wells, Corning, NY, USA), incubated with methanolic extracts added at different concentrations (0, 10, 25, 50 and 100 µg/mL) and dihydrorhodamine 123 (DHR, 1 µM, Cayman Chemical Company, Ann Arbor, MI, USA) and finally stimulated or not by phorbol 12-myristate 13-acetate (PMA, 1 µM, Sigma-Aldrich, Saint Louis, USA). Fluorescence readings (excitation/emission: 485/535 nm) in kinetics of rhodamine 123, which is the product of DHR 123 oxidation by ROS, were taken over 2 h every 10 min using a Fluoroskan Ascent FL^®^ apparatus (ThermoFisher Scientific, Illkirch, France) [65]. These experiments were repeated three times and the percentage of ROS production in the presence of the extracts was calculated. ROS production was calculated relative to 100% normalized control.

### 4.7. Effect of Extracts on Leukocyte Viability

Leukocytes preparations, obtained from the same donors, were placed in 96 well polystyrene plates incubated with methanolic extracts added at different concentrations (0, 10, 25, 50 and 100 µg/mL) and finally stimulated or not by PMA (1 µM). After 24 h of incubation at 37 °C under a 5% CO_2_ atmosphere, resazurin (Sigma-Aldrich) was added to each well at the concentration 25 µg/mL. Leukocytes viability and then cytotoxicity of the different extracts were monitored by fluorescence reading (excitation/emission: 530/590 nm) using a Fluoroskan Ascent FL^®^ instrument. The percentage of cell viability was calculated relative to 100% normalized control.

### 4.8. Data Analysis

Data were expressed as the mean ± SEM and represented at least three independent experiments. Statistical analysis and significance were measured using the paired, bilateral Student’s *t*-test. Differences at *p ≤* 0.05 (flagged as *) were considered statistically significant.

## 5. Conclusions

*V. madiensis* is one of the species of the *Vitex* genus which synthesizes ecdysteroids, steroid hormones from invertebrates which are involved in the regulation of moulting, development and reproduction. The biosynthesis of these compounds is carried out, in a variable way, at the level of root bark, barks and fruits. In addition, 20-hydroxyecdysone, vitexirone, and ajugasterone C are the major compounds found in *Vitex* genus among the ecdysteroids family. The extracts of leaves, trunk bark and fruits of *V. madiensis* were studied and showed a free radical scavenging and anti-inflammatory activities; moreover, both extracts did not show any cytotoxic effect on blood cells. The extracts of *C. febrifuga*, mainly contain organic acids, sugars and phenolic compounds, but the chemical composition is very dependent on the organ of the plant. Despite a significant cytotoxic effect on blood cells, the extracts of leaves, trunk bark and root bark of *C. febrifuga* showed a significant inhibition of DPPH.

This study allowed us to demonstrate that leaves and trunk bark extracts of *V. madiensis* present an anti-inflammatory activity by decreasing ROS production. As previously described, we supposed that the activity is due to the presence of ecdysteroids, but we will isolate all the major compounds of both extracts to evaluate the potential anti-inflammatory activity.

## Figures and Tables

**Figure 1 plants-12-00386-f001:**
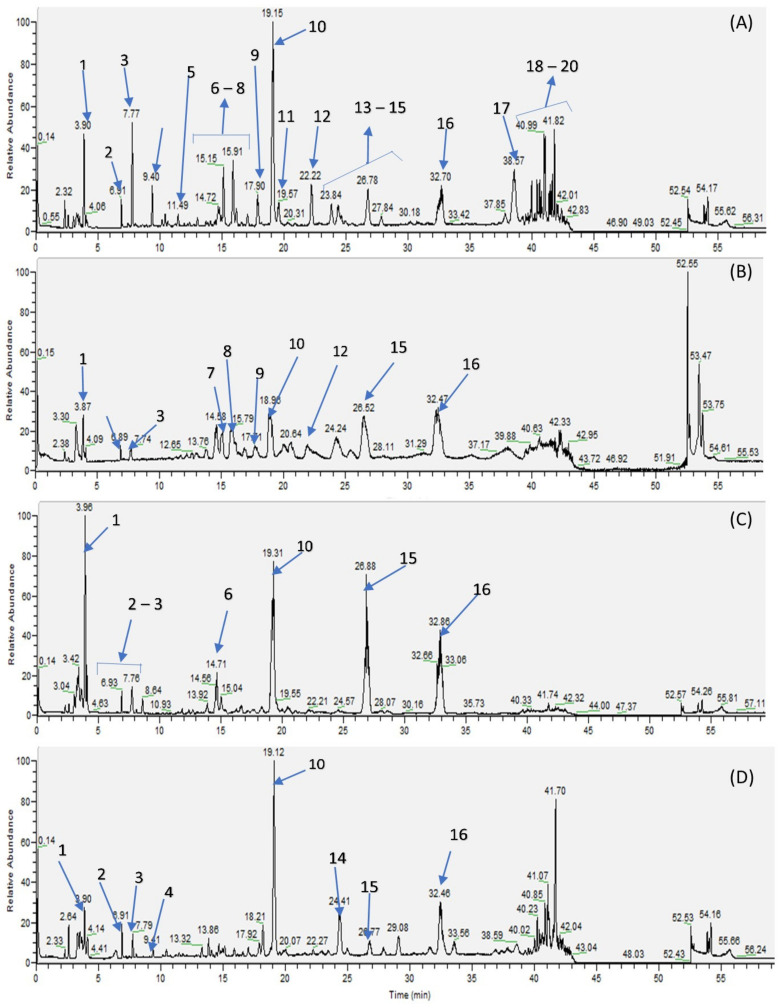
LC-MS chromatograms in negative mode of methanolic extracts of *V. madiensis* (**A**) leaves; (**B**) trunk bark; (**C**) root bark; (**D**) fruits.

**Figure 2 plants-12-00386-f002:**
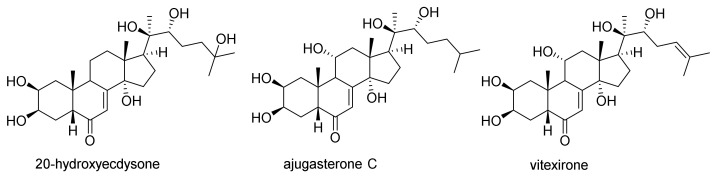
Structures of some ecdysteroids identified in *V. madiensis*.

**Figure 3 plants-12-00386-f003:**
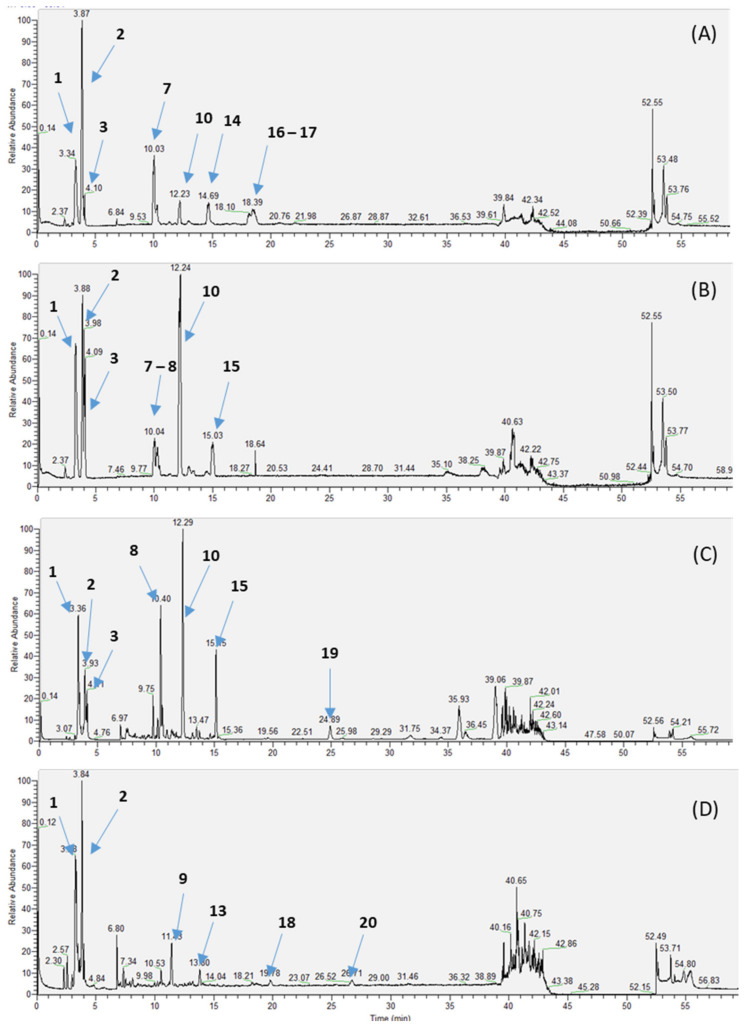
LC-MS chromatograms of *C. febrifuga* methanolic extracts: (**A**) leaves; (**B**) trunk bark; (**C**) root bark; (**D**) fruits.

**Figure 4 plants-12-00386-f004:**
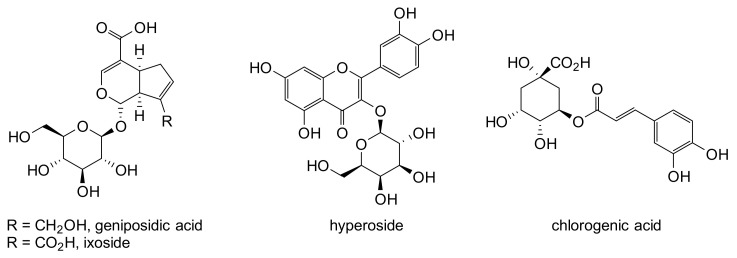
Structures of some compounds identified in *C. febrifuga*.

**Figure 5 plants-12-00386-f005:**
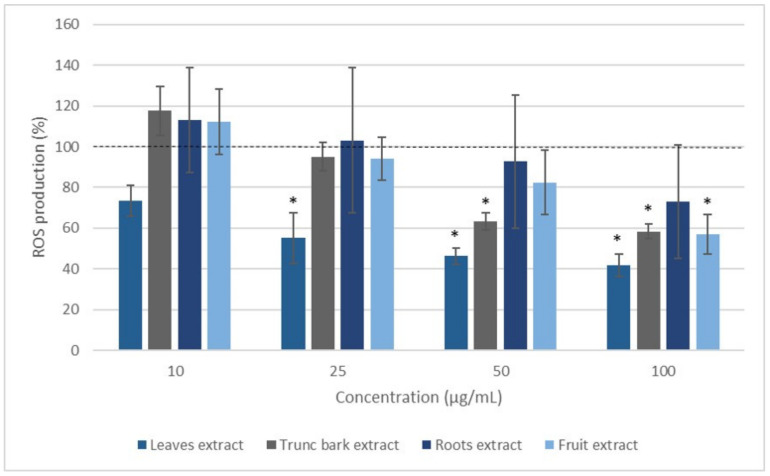
Effect of *Vitex madiensis* extracts on ROS production of blood leukocytes stimulated with PMA for 2 h. Values are expressed as percentage of the Control (cells incubated with PMA and without extract). * *p* < 0.05, compared with Control normalized as 100% (dotted line).

**Figure 6 plants-12-00386-f006:**
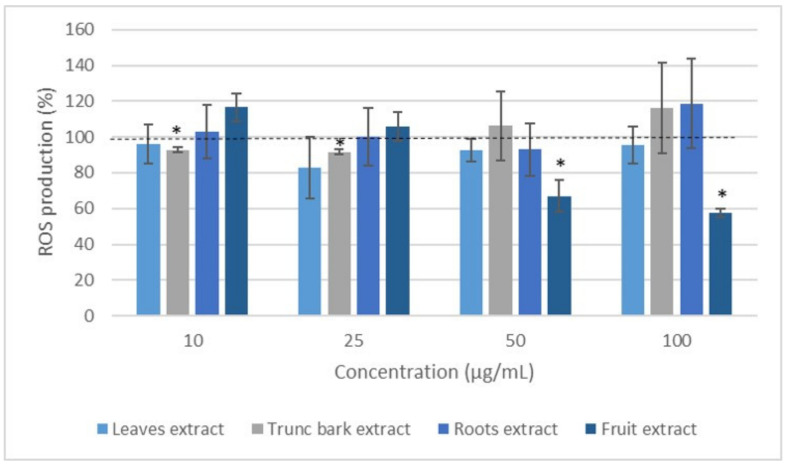
Effect of *Crossopteryx febrifuga* extracts on ROS production of blood leukocytes stimulated with PMA for 2 h. Values are expressed as percentage of the Control (cells incubated with PMA and without extract). * *p* < 0.05, compared with Control normalized as 100% (dotted line).

**Figure 7 plants-12-00386-f007:**
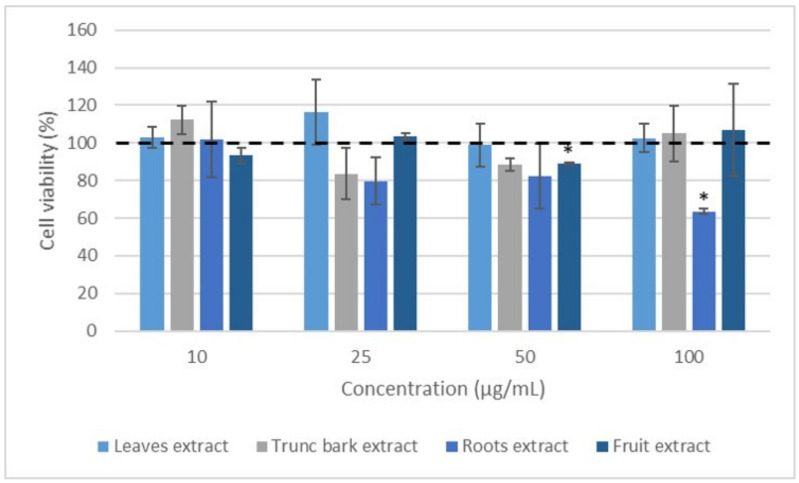
Effect of *Vitex madiensis* extracts on leukocyte viability. Blood cells were treated with the indicated concentrations of extracts, in the presence of PMA for 24 h. Data were shown as means ± SEM (Control = 100%), * *p* < 0.05 compared with Control.

**Figure 8 plants-12-00386-f008:**
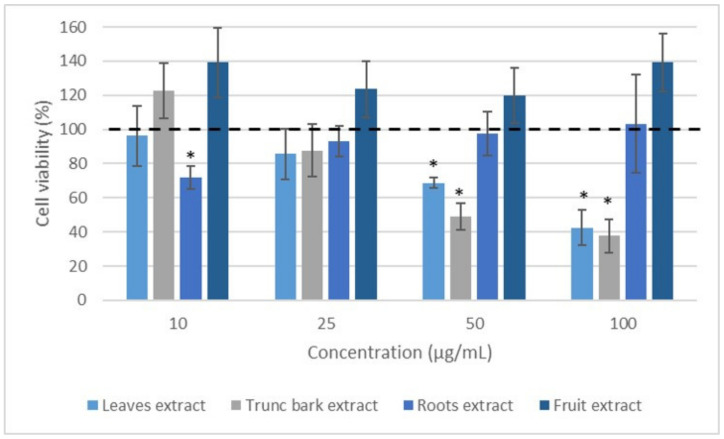
Effect of *Crossopteryx febrifuga* extracts on leukocyte viability. Blood cells were treated with the indicated concentrations of extracts, in the presence of PMA for 24 h. Data were shown as means ± SEM (Control = 100%), * *p* < 0.05 compared with Control.

**Table 1 plants-12-00386-t001:** Ecdysteroids and other compounds identified in extracts of leaves, trunk bark, root bark and fruits of *Vitex madiensis*.

N°	Compound	Tr (min)	Formula	M-H (m/z)	M + (m/z)	MS^2^ (m/z)	Le	TB	RB	F	Reference
1	Quinic acid	3.96	C_7_H_12_O_6_	191.0553		**191**/85/127/93/85	++	+	+++	+	Standard
2	Citric acid	6.91	C_6_H_8_O_7_	191.0189		**191**/129/11/87/85	+	+	+	+	Standard
3	1-oxo-eucommiol	7.76	C_9_H_14_O_5_	201.0757		**201**/109/139	++	+	+	+	[20]
4	5-Ethylidene-2-hydroxy-2-hydroxymethyl-3-methylhexanedioic acid	9.40	C_10_H_16_O_6_	231.0866		**231**/213/169/125/157/187/143	+	-	-	+	[21]
5	Protocatechuic acid	11.49	C_7_H_6_O_4_	153.0179		109/**153**/110	+	-	-	-	[22]
6	Vicenin-1	14.72	C_26_H_28_O_14_	563.1413		**563**/353/383/297/473/443/297/325/503/545	+	-	-	-	[23]
7	Homoorientin	15.15	C_21_H_20_O_11_	447.0928		429/357/327	+	+	+	-	Standard
8	Orientin	15.91	C_21_H_20_O_11_	447.0928		369/357/327/299	+	+	-	-	Standard
9	Vitexin	17.90	C_21_H_20_O_10_	431.0981			+	+	-	+	Standard
10	20-Hydroxyecdysone	19.15	C_27_H_44_O_7_	525.3065 *	481.3152	445/371/165/427/125/69/**481^#^**	+++	+	+++	+++	[24,25,26]
11	Luteolin-7-O-glucuronide	19.57	C_21_H_18_O_12_	461.0734		285	+	+	-	-	[27]
12	3,4-dicaffeoylquinic acid	22.22	C_25_H_24_O_12_	515.1201		353/191/179	+	+	+	+	Standard
13	Luteolin-4’-O-glucoside	23.84	C_21_H_20_O_11_	447.0939		285/**447**	+	+	-	-	Standard
14	Isovitexirone	24.41	C_27_H_42_O_7_	523.2917 *	479.2998	425/443/373/123/145/219/303/407/461/**479 ^#^**	+	+	-	+	[27]
15	Vitexirone	26.77	C_27_H_42_O_7_	523.2908 *	479.2998	69/443/425/299/109/407/**479**/461/281/311 **^#^**	+	+	+++	+	[28]
16	Ajugasterone C	32.70	C_27_H_44_O_7_	525.3061 *	481.3152	427/445/81/409/299/311/**481**/463/189 **^#^**	+	+	++	-	[29]
17	Luteolin	38.57	C_15_H_10_O_6_	285.0405			+	-	-	+	Standard
18	3,7-Dimethylquercetin	40.99	C_17_H_14_O_7_	329.0664		314/299/271/243	+	-	-	-	Standard
19	Eupatorin	41.82	C_18_H_16_O_7_	343.0824		328/298/270	+	-	-	-	[30]
20	Phytuberin	42.33	C_17_H_26_O_4_	293.1759		236/221/**293**	-	+	-	-	[31]

+: presence of the compound in the extract; ++: abundant compound; +++: very abundant compound; -: absence of the compound in the extract; L: leaves; TB: trunk bark; R: root bark; Fr: fruit. * (M+HCOO^-^); fragmented ions are highlighted in bold; # The MS^2^ analyses were performed in positive mode.

**Table 3 plants-12-00386-t003:** Inhibitory concentrations (IC_50_s) of extracts.

	*V. madiensis*				*C. febrifuga*				Ascorbic Acid
	L	TB	R	F	L	TB	R	F	
IC_50_ (µg/mL)	110	125	980	210	100	110	200	710	100

**Table 4 plants-12-00386-t004:** Yield of methanolic extraction of each part of plants (L: leaves, TB: trunk bark, RB: root bark, F: fruits).

	*Vitex madiensis*				*Crossopteryx febrifuga*			
Parts of plants	L	TB	RB	F	L	TB	RB	F
Mass of plants (g)	200	200	115	300	200	200	300	112
Mass of extract (g)	12	6	5.8	12	38	20	42	3.4
Yield (%)	6	3	5	4	19	10	14	3

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
