# Peer review of "Chemical Profile, Antioxidant and Anti-Inflammatory Potency of Extracts of Vitex madiensis Oliv. and Crossopteryx febrifuga (Afzel ex G. Don)"

_plants, 2023, doi:10.3390/plants12020386_

Round 1

Reviewer 1 Report

1. The title of section 4.8 should be different than 4.7.

2. Make botanical names of plants italics everywhere in manuscript

3. In Fig 2, authors have reported compounds with stereochemistry, I think normal LC-MS doesn't give stereochemistry of the compounds

4. Authors should provide a comparative mass spectrum of significant compounds (experimental with library search)

5. Anti-inflammatory is a broad term, so it would be better to write ROS inhibition activity

6. In the discussion part: Authors have written only about the biology part, a discussion of the chemistry part (identification of molecules by LC-MS ) should be added

Antioxidant activity of plants have been already reported in literature, authors should mention in discussion part in what extent this study is different than prev

Author Response

1.The title of section 4.8 should be different than 4.7.

We have merged the two sections.

  1. Make botanical names of plants italics everywhere in manuscript

We have modified the name of plants

  1. In Fig 2, authors have reported compounds with stereochemistry, I think normal LC-MS doesn't give stereochemistry of the compounds

We agreed that LC MS doesn’t give the stereochemistry of the compounds, we just estimated that the stereochemistry of the compounds described in literature on the same plants genus were established.

  1. Authors should provide a comparative mass spectrum of significant compounds (experimental with library search)

We added a supplementary part in which we included the comparison of the MS2 spectra of the extract and the library.

For ecdysteroids we don’t have the standards as described in the paper, we compared the MS2 fragments with the fragments described in literature

  1. Anti-inflammatory is a broad term, so it would be better to write ROS inhibition activity

Anti-inflammatory activity may indeed be a rather broad term and inhibition of ROS production is a parameter of this activity as there are others such as cytokines. We will therefore change this parameter to ROS production inhibition when it is judicious.

  1. In the discussion part: Authors have written only about the biology part, a discussion of the chemistry part (identification of molecules by LC-MS ) should be added.

We added a part about C. febrifuga in the discussion.

  1. Antioxidant activity of plants have been already reported in literature, authors should mention in discussion part in what extent this study is different than prev

The antioxidant activity of plants has been reported but the antioxidant activity of V. madiensis and C. febrifuga have not been described and the chemical profile was not clearly described. These two plants were chosen because they were not well described and because the ethnobotanic study carried out in Congo (Miss Boungou carry out an ethnobotanic survey among traditional healers) allowed us to confirm that these two plants were used to treat anti-inflammatory disease.

MS2 of the standard (from our library)

Orientin

MS2 of V. madiensis methanolic extract

MS2 of the standard (from our library)

Chlorogenic acid

MS2 of C. Febrifguga methanolic extract

MS2 of the standard (from our library)

Hyperoside

MS2 of C. Febrifguga methanolic extract

MS2 of the standard (from our library)

Isoquercitrine

MS2 of C. Febrifguga methanolic extract

MS2 of the standard (from our library)

Vitexin

MS2 of V. madiensis methanolic extract

MS2 of the standard (from our library)

Orientin

MS2 of V. madiensis methanolic extract

MS2 of the standard (from our library)

Chlorogenic acid

MS2 of C. Febrifguga methanolic extract

MS2 of the standard (from our library)

Hyperoside

MS2 of C. Febrifguga methanolic extract

MS2 of the standard (from our library)

Isoquercitrine

MS2 of C. Febrifguga methanolic extract

1.The title of section 4.8 should be different than 4.7.

We have merged the two sections.

  1. Make botanical names of plants italics everywhere in manuscript

We have modified the name of plants

  1. In Fig 2, authors have reported compounds with stereochemistry, I think normal LC-MS doesn't give stereochemistry of the compounds

We agreed that LC MS doesn’t give the stereochemistry of the compounds, we just estimated that the stereochemistry of the compounds described in literature on the same plants genus were established.

  1. Authors should provide a comparative mass spectrum of significant compounds (experimental with library search)

We added a supplementary part in which we included the comparison of the MS2 spectra of the extract and the library.

For ecdysteroids we don’t have the standards as described in the paper, we compared the MS2 fragments with the fragments described in literature

  1. Anti-inflammatory is a broad term, so it would be better to write ROS inhibition activity

Anti-inflammatory activity may indeed be a rather broad term and inhibition of ROS production is a parameter of this activity as there are others such as cytokines. We will therefore change this parameter to ROS production inhibition when it is judicious.

  1. In the discussion part: Authors have written only about the biology part, a discussion of the chemistry part (identification of molecules by LC-MS ) should be added.

We added a part about C. febrifuga in the discussion.

  1. Antioxidant activity of plants have been already reported in literature, authors should mention in discussion part in what extent this study is different than prev

The antioxidant activity of plants has been reported but the antioxidant activity of V. madiensis and C. febrifuga have not been described and the chemical profile was not clearly described. These two plants were chosen because they were not well described and because the ethnobotanic study carried out in Congo (Miss Boungou carry out an ethnobotanic survey among traditional healers) allowed us to confirm that these two plants were used to treat anti-inflammatory disease.

Reviewer 2 Report

Accepted in present form

Author Response

The reviewer 2 accepted the manuscript in the present form

Reviewer 3 Report

The paper of Boungou-Tsona et al. aimed to study the chemical composition and bioactivity of Vitex madiensis and Crossopteryx febrifuga, the African plants with ethnopharmacological purpose but lack of knowledge. The results are generally new, but the chaotic building of the draft and the obvious errors does not allow to recommend it for the further publication. The followed problem points must necessarily be rectified.

-          The positive ionization mode is very useful for ecdysteroid detection. You are claiming that only amino acids were detected in positive mode but [M+H]+ data of ecdysteroids included in Table 1. Could you explain your statement.

-         Table 1.  It is unclear, how to read column 7 (MS2 data). The daughter ions are listed here not in order from larger to smaller. The meaning of bold numbers is unclear too. The parent ion is not marked.

-          Tentatively identified compounds should me indicated by a special sign.

-          The basic identification of 20-hydroxyecdysone (20E) was made because of MS2 spectrum, not reference standard. The MS data includes the following ions 481 [M+H], 445 [(M+H)-2H2O], 427 [(M+H)-3H2O], 371 [(M+H)-C4H10O-2H2O], 165, 125, 99. The same fragments were found for some isomeric to 20E ecdysteroids as 22-deoxy-20,21-dihydroxyecdysone, 22-deoxyintegristerone A, 5-hydroxyecdysone, 11-hydroxyecdysone, inokosterone, etc. How could you differentiate 20E from isomeric compounds? The same remark for compounds 14-16.

-          Compound 13 (Luteolin-4'-O-glucoside) has two fragments in MS spectrum as 447 [M-H] and 285 [(M-H)-hex]. How could you differentiate 4'-O-glucoside from 7-O-glucoside or 3'-O-glucoside have the same spectral pattern?

-          How can you determined it ? " ++ : abundant compound ; +++ : very abundant compound " The area of MS peak on chromatogram is not a criterion of abundancy.

-          Lines 95, 100, etc. When you refer to fatty acids, what compounds you mean? I can't find any fatty acid in Table 1.

-          Line 97. Only one dicaffeoylquinic acid was found, not acids.

-          Figure 2. Structures of ecdysteroids and other compounds identified in V. madiensis. What other compounds did you mean?

-          Can't see Table 2.

-          Table 3. How could you differentiate gluconic acid from other 15 isomers of 2,3,4,5,6-pentahydroxyhexanoic acid have the same MS pattern and unseparatable in your HPLC conditions?

-          Compound 17. Isoquercetin means isoquercitrin?

-          Figure 4. Structures of major compounds identified in C. febrifuga. How did you determine these four compounds as major? quantitatively? Where is the data?

-          Table 3. Want ++, +++ means?

-          The low DPPH value of ascorbic acid (100 μg/mL). I understand that the results of DPPH value determination can vary in wide range for various authors and methods used, but IC50 = 100 μg/mL for ascorbic acid (a strongest reducer, with "traditional" level of IC50 value 0.5-10 μg/mL) is too low. There is a sheer discrepancy here that needs explanation.

- The results of bioactivity study in Sections 2.3 and 2.4 need additional information about reference standard potential. Otherwise, it is not possible to conclude the effectiveness of studied extracts.

Author Response

-The positive ionization mode is very useful for ecdysteroid detection. You are claiming that only amino acids were detected in positive mode but [M+H]+ data of ecdysteroids included in Table 1. Could you explain your statement :

We agree that this statement is not really clear. We changed this sentence in the text. We performed analyses in both modes (positive and negative). The ecdysteroids are also ionized in positive mode and in the negative mode but the negative mode give almost all the chemical profile including ecdysteroids, flavonoides, phenols. We didn’t want to add the mass spectra in positive mode, we think that 4 chromatogramms in a “letter” should probably be enough.

- Table 1.  It is unclear, how to read column 7 (MS2 data). The daughter ions are listed here not in order from larger to smaller. The meaning of bold numbers is unclear too. The parent ion is not marked.

We are used to list the daughter ions depending on the abundance of the ion. I think most of the authors present the results of MS2 in this order.

-Tentatively identified compounds should me indicated by a special sign.

We specified in the table the compounds that have been identified with a standard reference (internal library) and the compounds that have been identified by comparing with fragments found in literature (a reference is mentioned for all these presumed compounds).

So, we don’t think it is necessary to add a special signal.

-          The basic identification of 20-hydroxyecdysone (20E) was made because of MS2 spectrum, not reference standard. The MS data includes the following ions 481 [M+H], 445 [(M+H)-2H2O], 427 [(M+H)-3H2O], 371 [(M+H)-C4H10O-2H2O], 165, 125, 99. The same fragments were found for some isomeric to 20E ecdysteroids as 22-deoxy-20,21-dihydroxyecdysone, 22-deoxyintegristerone A, 5-hydroxyecdysone, 11-hydroxyecdysone, inokosterone, etc. How could you differentiate 20E from isomeric compounds? The same remark for compounds 14-16.

We agreed that various isomers of 20E ecdysone are described but we hope that the results obtained previously on V. Madiensis are right. We just confirmed the results of previous studies, except for pterosterone, for which we make a hypothesis (this compound has not been yet described in V. Madiensis.

-          Compound 13 (Luteolin-4'-O-glucoside) has two fragments in MS spectrum as 447 [M-H] and 285 [(M-H)-hex]. How could you differentiate 4'-O-glucoside from 7-O-glucoside or 3'-O-glucoside have the same spectral pattern?

We analyzed three different isomers (luteolin-3, 4 and 7- glucoside) using the same LC method. As we don’t have the same retention time we can affirm that the compound 13 is luteolin-4-O-guloside (and not the 4’, it was a mistake).

-          How can you determined it ? " ++ : abundant compound ; +++ : very abundant compound " The area of MS peak on chromatogram is not a criterion of abundancy.

The ++ and +++ is just a comparison of the abundance between the four parts of the plants as the analyses were all performed with the same concentration of extract.

-          Lines 95, 100, etc. When you refer to fatty acids, what compounds you mean? I can't find any fatty acid in Table 1.

It is a mistake. It is not fatty acids but amino acids. We didn’t mention the few amino acids detected in the extracts as we focused on antioxidant compounds.

-          Line 97. Only one dicaffeoylquinic acid was found, not acids.

We have modified the sentence

-          Figure 2. Structures of ecdysteroids and other compounds identified in V. madiensis. What other compounds did you mean?

It is a mistake. We have modified the title.

-          Can't see Table 2.

We have modified the table number

-          Table 3. How could you differentiate gluconic acid from other 15 isomers of 2,3,4,5,6-pentahydroxyhexanoic acid have the same MS pattern and unseparatable in your HPLC conditions?

We have changed gluconic acid by uronic acid. We agree that it is difficult to identify the nature of the sugar.

-          Compound 17. Isoquercetin means isoquercitrin?

We have modified the name

-          Figure 4. Structures of major compounds identified in C. febrifuga. How did you determine these four compounds as major? quantitatively? Where is the data?

We have modified the title. We effectively couldn’t say that these four compounds as the major compounds as we only performed an LC/MS analysis.

-          Table 3. Want ++, +++ means?

The ++ and +++ is just a comparison of the abundance between the four parts of the plants as the analyses were all performed with the same concentration of extract.

-          The low DPPH value of ascorbic acid (100 μg/mL). I understand that the results of DPPH value determination can vary in wide range for various authors and methods used, but IC50 = 100 μg/mL for ascorbic acid (a strongest reducer, with "traditional" level of IC50 value 0.5-10 μg/mL) is too low. There is a sheer discrepancy here that needs explanation.

The IC50 was determined at a concentration so that the activity of the extract could be compared to the ascorbic acid IC50.

Some authors published an IC50 of acid ascorbic around 60 μg/mL.

Volatile Fraction Composition and Total Phenolic and Flavonoid Contents of Elionurus hensii—Antioxidant Activities of Essential Oils and Solvent Extracts Yin Yang , Marie-Cécile De Cian , Samuel Nsikabaka , Pierre Tomi , Thomas Silou , Jean Costa and Julien Paolini, Nat Prod Comm. 2013. DOI: 10.1177/1934578X1300800528

- The results of bioactivity study in Sections 2.3 and 2.4 need additional information about reference standard potential. Otherwise, it is not possible to conclude the effectiveness of studied extracts.

ROS production by leukocytes was studied in the presence and absence of extract. Cells without extract serve as a control condition. This experimental control had 100% of ROS production. It is our practice to publish in this way. This was the case for example in the publication In Vitro Anti-Inflammatory and Immunomodulatory Activities of an Extract from the Roots of Bupleurum rotundifolium (Medicines 2019, 6, 101; doi:10.3390/medicines6040101)

Other authors also published this way: for example in Foods 2022, 11(13), 1832; https://doi.org/10.3390/foods11131832

Here, in the graphs of sections 2.3 and 2.4 this control condition is represented by the dotted line which corresponds to a ROS production equal to 100%.

We study the production of intracellular ROS in human cells from different subjects. Thus, a decrease of nearly 60% compared to the control corresponds to an effective decrease in intracellular production.

Cells 2021, 10, 2691. https://doi.org/10.3390/cells10102691

Round 2

Reviewer 3 Report

In spite of the reforms, the question remains how precise mass-spectrometric structure determination was done for ecdysteroids and flavone glycosides without standards. There are no any answers to the questions:

- The basic identification of 20-hydroxyecdysone (20E) was made because of MS2 spectrum, not reference standard. The MS data includes the following ions 481 [M+H], 445 [(M+H)-2H2O], 427 [(M+H)-3H2O], 371 [(M+H)-C4H10O-2H2O], 165, 125, 99. The same fragments were found for some isomeric to 20E ecdysteroids as 22-deoxy-20,21-dihydroxyecdysone, 22-deoxyintegristerone A, 5-hydroxyecdysone, 11-hydroxyecdysone, inokosterone, etc. How could you differentiate 20E from isomeric compounds? The same remark for compounds 14-16.

- Compound 13 (Luteolin-4'-O-glucoside) has two fragments in MS spectrum as 447 [M-H] and 285 [(M-H)-hex]. How could you differentiate 4'-O-glucoside from 7-O-glucoside or 3'-O-glucoside have the same spectral pattern?

Your conclusions are looks as tentative not precise.

Author Response

- The basic identification of 20-hydroxyecdysone (20E) was made because of MS2 spectrum, not reference standard. The MS data includes the following ions 481 [M+H], 445 [(M+H)-2H2O], 427 [(M+H)-3H2O], 371 [(M+H)-C4H10O-2H2O], 165, 125, 99. The same fragments were found for some isomeric to 20E ecdysteroids as 22-deoxy-20,21-dihydroxyecdysone, 22-deoxyintegristerone A, 5-hydroxyecdysone, 11-hydroxyecdysone, inokosterone, etc. How could you differentiate 20E from isomeric compounds? The same remark for compounds 14-16.

We identified the structure of the ecdsteroids first because they have been described in Vitex genus.

Concerning the position of the hydroxyl groups we are almost sure that we identified 20E because the fragments in MS2 depend on the position of the hydroxyl group.

For 20E we added a publication (Zhou, J.; Qi, Y.; Hou, Y.; Zhao, J.; Li, Y.; Xue, X.; Wu, L.Zhang, J.; Chen, F. Quantitative determination of juvenile hormone III and 20-hydroxyecdysone in queen larvae and drone pupae of Apis mellifera by ultrasonic-assisted extraction and liquid chromatography with electrospray ionization tandem mass spectrometry, J. Chromatogr. B 2011, 879, 2533-2541) explaining the fragment pathway. The fragments 445 and 371 are characteristic of 20E:

  • only three hydroxyl groups in the 4 ring skeleton
  • position of the hydroxyl group on the side chain.

For us it is the same explanation for other ecdysteroids.

- Compound 13 (Luteolin-4'-O-glucoside) has two fragments in MS spectrum as 447 [M-H] and 285 [(M-H)-hex]. How could you differentiate 4'-O-glucoside from 7-O-glucoside or 3'-O-glucoside have the same spectral pattern?

We carried out the LC/MS analysis of both standard of Luteolin-4'-O-glucoside and Luteolin-7-O-glucoside and showed that these two isomers had a different retention time : respectively 23,8 and 18,9 minutes. As the compound identified in the extract had a retention tile of 23, 84 minutes (of course with the same column and the same method) we conclude that the compound presents in the extract is Luteolin-4'-O-glucoside.
